# An Integrated Approach to Workplace Mental Health: A Scoping Review of Instruments That Can Assist Organizations with Implementation

**DOI:** 10.3390/ijerph20021192

**Published:** 2023-01-09

**Authors:** Adam Nebbs, Angela Martin, Amanda Neil, Sarah Dawkins, Jessica Roydhouse

**Affiliations:** 1Menzies Institute for Medical Research, College of Health and Medicine, University of Tasmania, Hobart 7000, Australia; 2School of Psychological Sciences, College of Health and Medicine, University of Tasmania, Hobart 7000, Australia

**Keywords:** workplace mental health, instruments, implementation research, surveys, questionnaires

## Abstract

This study aimed to identify instruments that may assist organizations with implementing an integrated approach to workplace mental health using three activities from the knowledge to action (KTA) framework. A scoping review of published and grey literature, supported by stakeholder (business end-user and researcher) consultation, identified work-specific instruments that were relevant to at least one of the three domains of an integrated approach to workplace mental health: ‘prevent harm’, ‘promote the positive’, and ‘respond to problems’. A total of 207 instruments were located, and 109 instruments met eligibility criteria. 10 instruments were located that were relevant to multiple domains, however most instruments (n = 72) were relevant to the ‘prevent harm’ domain. Instruments relevant to the ‘promote the positive’ (n = 14) and ‘respond to problems’ (n = 13) domains were limited. Most instruments found were suitable for the ‘monitor, review and improve’ KTA activity. Further development of instruments that can assist with ‘promote the positive’ and ‘respond to problems’ strategies are required, specifically those instruments that can assist organizations with the ‘identify gaps and opportunities’ and ‘identify priorities and design new/enhanced interventions’ KTA activities.

## 1. Introduction

Mental health concerns are common in the working population [1]. An estimated 15% of working age adults have a mental disorder at any one time [2]. It is estimated that common mental disorders, such as depression and anxiety, cost the global economy USD 1 trillion each year, primarily due to lost productivity [3]. The issue of worker mental health has, thus, been given significant attention, in recent decades, by policy makers, tripartite agencies, social partners (workers, governments and employers), and academics [4]. To address worker mental health, employers should focus on creating and maintaining a mentally healthy workplace. A mentally healthy workplace has been conceptualized as “one in which risk factors are acknowledged and appropriate action taken to minimize their potential negative impact on an individual’s mental health. At the same time protective or resilience factors are fostered and maximized” [5]. One approach to creating a mentally healthy workplace is to apply “an integrated approach to workplace mental health” [6].

An integrated approach to workplace mental health is a principles-based, evidence informed framework that guides the protection and promotion of worker mental health through three overlapping domains of action (see Figure 1): ‘prevent harm’ (protecting mental health by reducing work-related risk factors), ‘promote the positive (promoting mental health by developing the positive aspects of work as well as worker strengths and positive capacities), and ‘respond to problems’ (addressing mental health problems among working people regardless of cause) [6,7]. Interventions, according to an integrated approach, should be directed at both the workplace and the worker, and combine primary, secondary, and tertiary prevention methods to be comprehensive and systems focused [8]. Systematic reviews of job stress prevention have shown that a systems-level approach is the most effective [9,10,11]. Positive and strength-based approaches that promote worker mental health are expected to complement these efforts, giving rise to additional benefits in work environments with high levels of mental health literacy and workplace support for those experiencing a mental illness [6,7]. However, translating integrated approaches from research to practice is still an emerging area [6,7]. Currently, the only knowledge translation strategy that has been used for implementing an integrated approach is the knowledge to action (KTA) framework [12].

### 1.1. An Overview of an Integrated Approach to Workplace Mental Health

The ‘prevent harm’ domain is largely based on principles and evidence from the fields of public health and organizational psychology [6,7]. The most comprehensive strategy for ‘preventing harm’, according to an integrated approach, combines primary, secondary, and tertiary interventions aimed at both the individual and organizational levels [6,7]. The ‘promote the positive’ domain of an integrated approach is predominately based on principles and evidence from the fields of positive organizational behavior, positive organizational scholarship, and positive psychology, and is the newest area of research of the three domains [13,14,15,16,17]. Interventions focus on strength-based methods, aiming to identify and enhance work-related factors that promote subjective well-being or positive mental health [18]. The ‘respond to problems’ domain of an integrated approach is predominantly based on principles and evidence taken from an illness or medical perspective [6,7]. Interventions within the ‘respond to problems’ domain focus on building mental health literacy and providing psychoeducation, with strategies aligned to support provision, promoting professional help seeking, reducing stigma, and increasing ‘safe’ disclosure in the workplace to allow for ‘reasonable accommodations’ to be implemented for those experiencing mental illness [6,7]. The ‘respond to problems’ domain also includes facilitating successful return to work outcomes after a mental illness related absence [6,7].

An integrated approach to workplace mental health provides a framework that organizations can follow to create a mentally healthy workplace. However, there is no single ‘recipe’ for implementing an integrated approach as each organization is unique and will require a strategy that is tailored and context appropriate [6,7]. Detailed implementation research has been called for, to help translate an integrated approach from research into practice [6,7].

### 1.2. Implementation of an Integrated Approach to Workplace Mental Health

Knowledge translation guidance in this area outlines a framework that organizations can take when implementing an integrated approach to workplace mental health. An unpublished knowledge translation guide, developed by Beyond Blue, outlines six activities that organizations can take when implementing an integrated approach [12]. The Beyond Blue guide was developed using the KTA framework that organizations can draw on when adapting an integrated approach to their local context [12,19]. The illustration below (Figure 2) outlines the high-level KTA framework.

Three of the six activities contained within Beyond Blue’s KTA framework relate to ‘identifying gaps and opportunities’ (activity 3), ‘identifying priorities, and designing new/enhanced interventions’ (activity 4), and creating a system to ‘monitor, review and improve’ (activity 6). To assist organizations seeking to implement an integrated approach to workplace mental health, the current study aimed to locate and catalogue extant instruments that can support these three KTA activities.

### 1.3. Aim of the Present Study

An integrated approach to workplace mental health is still an emerging area of research, thus, the present study poses an exploratory research question:


*“What instruments are available to assist organizations with the process of implementing an integrated approach to workplace mental health?”*


## 2. Materials and Methods

### 2.1. Study Design and Protocol

A scoping review methodology was selected as the body of literature and resources under inspection are large and heterogenous [20]. A scoping review approach also enabled us to engage with each stage of the review in a reflexive way, as familiarity with the literature and resources increased [20]. The scoping review was conducted according to the Preferred Reporting Items for Systematic reviews and Meta-Analyses extension for Scoping Reviews (PRISMA-ScR) statement [21]. Our protocol was registered prospectively with the Open Science Framework on 5 December 2020 (https://osf.io/pe9zx/ accessed on 28 July 2021). Due to our research methods being complex, and consisting of multiple phases within different steps, our methods have been summarized in Figure 3.

### 2.2. Operationalizing ‘Workplace Mental Health Instruments’

The literature under review was complex and significant, requiring a broad search methodology. A scoping review was considered useful in mapping the existing literature and determining the boundaries of what instruments were available to assist organizations with the process of implementing an integrated approach to workplace mental health [20].

For the purposes of this paper, the word ‘instrument’ will be used when referring to measures, scales, questionnaires, tests, and inventories, among other related measurement terms [22]. The word ‘item’ will be used when referring to the questions contained within each instrument. A ‘workplace mental health instrument’ was operationalized using the associated ‘prevent harm’, ‘promote the positive’, and ‘respond to problems’ variables discussed in existing integrated approach research [6,7]. An instrument was operationalized as being suitable for one or more of the three KTA activities based on the description written in the papers that explain the KTA framework [12,19]. It was also decided to review whether the located instruments had any information or resources on the instrument’s development. An instrument was classified as having evidence of development based on the presence or absence of development information or resources, sourced through a review of specific scientific databases, grey literature, and stakeholder consultation.

The details of how a ‘workplace mental health instrument’ was operationalized, how that instrument was assessed against one or more of three KTA activities, and what was required for an instrument to be assessed as providing development resources is explained further in Table 1 and Table 2.

To address the research question, we took a two-fold approach. First, the data were recorded in the data charting form and presented in two tables; the first, a list of the included instruments as they sat against the domains of an integrated approach to workplace mental health, the author and year, the country of origin, and a brief description of the instrument. Second, to provide an overall picture, we summarized the number of instruments that were mapped to one or more domains of an integrated approach, the KTA activities, and that included development resources. Taken together, this provided a catalogue of instruments as well as an overall view of what was available for assisting with the process of implementing an integrated approach to workplace mental health.

### 2.3. Eligibility Criteria

The inclusion and exclusion criteria utilized throughout the scoping review are detailed in Table 1. It was necessary to add an inclusion criterion around the availability of an instrument as some instruments did not have publications or resources that were easily available for content review [23]. It was also decided that any instruments that could only be reviewed upon their purchase would be excluded from further review unless the commercial entity was prepared to provide the instrument to the researchers for this review. Any commercial instruments would be signposted. A language restriction was also added, with English-language only resources selected for review.

### 2.4. Information Sources and Search

Before commencing the review, an initial search of Google and Google Scholar was completed to identify relevant literature to refine the search strategy. The key words located in several of the relevant research articles were then used to formulate a search string for the formal review. These included “Job Stress”, “Psychosocial Risk”, “Mental Health”, “Psychological Health”, “Risk Assessment”, and “Surveys and Questionnaires”. This approach was in keeping with scoping review guidelines that recommend text words and index terms are reviewed after completing a limited search of relevant databases [20]. A three-phase approach was employed for the full review, encompassing academic, grey literature and stakeholder consultation.

Searches were initially completed between December 2020 and June 2021 by A.N. (first author). However, this search was later expanded from June 2021 until September 2022, yielding an additional 11 instruments for inclusion. The first phase of the search was a comprehensive review of relevant databases (Medline, PsychINFO, Web of Science, SCOPUS, and Google Scholar). The search terms are recorded in Appendix A.

Each database search string was updated after a review of the text and index terms found in the key words section of each article that were included for title and abstract screening purposes. The first 1250 entries in each database were reviewed, with a total of 5000 articles reviewed across the four scientific databases to reach data saturation. The decision was made to limit the Google Scholar search, having noted the lack of relevant entries in the initial search of Google Scholar, to the first 200 entries. The reference lists of papers provided for screening were then hand searched to identify additional relevant articles.

The second phase involved a comprehensive search of grey literature using the Google Advanced function to customize the search and allow for a refined and targeted search strategy [24]. Customizations included site limitations, language restrictions and limitation of searches to a particular filetype. In this instance, English-only pages were chosen, with the filetype as “pdf”, and the site domains as “gov”, “gov.au”, “org”, “org.uk”, and “edu”. The first search of each string also looked at the “inURL” only option which screens URLs for the keywords selected. The reference or resource lists in any of the included grey literature were also hand searched to identify any additional resources.

The third phase was the creation of a stakeholder group (n = 26) that were recruited through a LinkedIn post, or contacted directly through LinkedIn, who could contribute knowledge of any additional instruments or suggest further resources that could be reviewed to identify any potentially relevant instruments. A targeted website review was also conducted on 27 different websites, including the Centre for Disease Control (CDC), World Health Organization (WHO), International Labor Organization (ILO), and Institution of Occupational Safety and Health (IOSH). The main psychological assessment publishing houses’ catalogues (PAR, Thames Valley Test Company, SHL Psychological Corporation, Central Test, Measure Chest, and Mind Garden) were also reviewed for any relevant materials. These websites were chosen after consultation with the stakeholder group.

### 2.5. Selection of Sources of Evidence

The scientific articles included for screening were managed using the systematic review system, Covidence (v2601). All other entries, including the grey literature, Google Scholar articles, websites, stakeholder advice, and relevant references were collated via separate purpose-built Excel spreadsheets. The articles and resources included for title and abstract screening were double screened by a second reviewer, A.M., with any conflicts reconciled through discussion or joint consultation with a third reviewer, J.R.

### 2.6. Data Charting Process/Data Items

A data charting form was developed for recording the data used to answer our research question, according to our specific definitions (Table 2). This form changed significantly from the draft version included in the protocol as a better understanding of which data needed to be extracted to answer our research question was formed. The data charting form consisted of three coding categories. The first category was about determining whether an instrument could assess one or all of the domains of an integrated approach to workplace mental health. The second category was about determining whether an instrument might assist organizations to ‘identify gaps and opportunities’, ‘identify priorities and design new/enhanced interventions’, ‘monitor, review and improve’ or for multiple purposes. The third category was about determining whether there were any resources or information available on the development of an instrument.

The decision was made to review whether an instrument, examined at the scale level, could be used to assess one or all of the domains of an integrated approach to workplace mental health, as defined in our eligibility criteria (see Table 1 and Table 2). Interpretation of the scales, within the included instruments, was difficult as the instruments were originally developed to assess other constructs [22]. Each of the scales included in the instrument was coded using the operationalized variables of ‘prevent harm’, ‘promote the positive’, and ‘respond to problems’ (see Table 2) [6,7]. In reviewing an instrument, it was decided that if the majority of the scales aided in the assessment of one domain (such as ‘prevent harm’) that the instrument would be considered most relevant to that domain. However, if there was an even number of scales relevant to more than one of the domains, the instrument was considered relevant to both domains. Any scales that captured an individual’s mental health outcomes were coded as “NA” as our focus was on the work-related factors that were associated with these outcomes. The two reviewers (A.N. and A.M.) independently charted the data with an interrater reliability of 100%.

## 3. Results

### 3.1. Synthesis of Results

The three phases of our search strategy identified 388 resources that were eligible for title and abstract screening, with 73 resources identified for full text review (Figure 4). A total of 221 instruments were identified through these 73 resources and were then double screened for eligibility (A.N. and A.M). Of these, 109 instruments were deemed eligible and included in the paper (Table 3).

### 3.2. Results and Synthesis of Individual Sources of Evidence

A summary of the number of instruments that were mapped to one or more domains of an integrated approach and their suitability to one or more of the KTA activities are captured in Table 4.

***Alignment with integrated approach domains: determining whether an instrument can assess ‘prevent harm’, ‘promote the positive’, ‘respond to problems’ or all three domains***: The FlourishDX and Guarding Minds at Work resources were the only instruments located that were relevant to all three domains of an integrated approach to workplace mental health [25,26]. A total of 8% of instruments were relevant to multiple domains, being ‘prevent harm’ and ‘promote the positive’ (n = 6) [27,28,29,30,31,32] and ‘prevent harm’ and ‘respond to problems’ (n = 2) [33,34]. No instruments, aside from FlourishDX and Guarding Minds at Work, were relevant to the ‘promote the positive’ and ‘respond to problems’ domains together. A total of 66% of instruments (n = 72) were relevant to the ‘prevent harm’ domain only, with 13% of instruments (n = 14) only relevant to the ‘promote the positive’ domain, and 12% of instruments (n = 13) to the ‘respond to problems’ domain. Most of the instruments relevant to the ‘prevent harm’ domain assessed psychosocial hazards or work-related stress factors, with some instruments assessing organizational life [35,67,89], job design [55,61], and management behaviors [59,73]. Workplace well-being and meaningful work were the most common factors assessed by the instruments relevant to the ‘promote the positive’ domain, with some instruments designed to assess psychological capital [112], social capital [118], and belonging [120]. In contrast, the assessment of employee knowledge, understanding and attitudes towards workplace mental health were the most common factors in instruments relevant to the ‘respond to problems’ domain, with some instruments that assessed manager attitudes [122] and return to work obstacles [126].

***KTA activities: determining whether an instrument may be suitable to assist organizations to ‘identify gaps and opportunities’, ‘identify priorities and design new/enhanced interventions’, ‘monitor, review and improve’ or for multiple purposes***: A total of 4% of instruments (n = 4) were found that may assist organizations in completing all three of the KTA activities. The instruments that may assist with all three KTA activities were FlourishDX, Guarding Minds at Work, the Workplace Psychological Safety Index, and the WorkPlace Wellbeing Insights Survey [25,26,32,103]. The majority of instruments (n = 90) found may assist organizations with the ‘monitor, review and improve’ activity, with most of these instruments relevant to the ‘prevent harm’ domain. A total of 14% of instruments (n = 15) found may assist organizations to ‘identify gaps and opportunities’ and ‘identify priorities and design new/enhanced interventions’.

***Accessibility of information about the instruments themselves: resources/information on the development of an instrument:*** A total of 77% (n = 84) of instruments had resources or information on how the instrument was developed. The majority (n = 76) of the resources on development, located through scientific databases, grey literature, and stakeholder consultation, were research articles. The remaining (n = 8) resources located were government and industry reports. A total of 23% (n = 25) of instruments had no resources or information on how the instrument was developed. A total of 81% (n = 59) of instruments relevant to the ‘prevent harm’ domain had resources or information on how the instrument was developed. A total of 92% (n = 13) of ‘promote the positive’ instruments, and 42% (n = 3) of ‘respond to problems’ instruments had resources or information on how the instrument was developed. A total of 95% (n = 60) of instruments sourced through scientific databases and references had information about the development of the instrument. A total of 54% of the instruments located through grey literature had resources or information on the development of an instrument, however, most of these instruments were located through a researcher-developed fact sheet [132]. A total of 43% (n = 7) of the instruments sourced through stakeholder consultation had information and resources on how the instrument was developed.

## 4. Discussion

This paper identified 109 workplace mental health instruments that are available to assist organizations interested in implementing an integrated approach to workplace mental health using the KTA framework. A total of 10 instruments were located that were relevant to multiple domains of an integrated approach. Most of the instruments located may be suitable for the ‘monitor, review and improve’ activity specific to the ‘prevent harm’ domain. Instruments relevant to the ‘promote the positive’ domain were largely absent and those found were primarily designed for the ‘monitor, review and improve’ activity and aimed at individual-level well-being factors. Workplace mental health instruments relevant to the ‘respond to problems’ domain were the least common in this review with most of those located suitable for ‘identifying gaps and opportunities’ and/or ‘identifying priorities and designing new/enhanced interventions’. Most instruments located in this review had resources and information on their development.

The only instruments relevant to all three domains of an integrated approach were FlourishDX and Guarding Minds at Work [25,26]. FlourishDX is a commercial suite of resources that includes a work design survey, risk assessment and tools for implementing the ISO 45003 psychological health and safety at work standard [25]. The Guarding Minds at Work instrument is based on the 13 psychosocial factors contained within the Canadian National Standard for Psychological Health and Safety in the Workplace [26]. The Canadian National Standard exemplifies one integrated approach that has already been employed in policy and practice in Organization for Economic Co-operation and Development (OECD) countries [6,7,133]. Six other instruments were relevant to the ‘prevent harm’ and ‘promote the positive’ domains, with some crossover observed during the mapping exercise [27,28,29,30,31,32]. It was noticeable, when reviewing these instruments, that a scale for assessing a psychosocial hazard, such as poor social support, may also be suitable for assessing the presence of a positive factor, such as social capital. Connections between the ‘prevent harm’ and ‘promote the positive’ domains have already been observed in previous integrated approach research [6,7]. However, a similar crossover was not observed for instruments that were relevant to both the ‘prevent harm’ and ‘respond to problems’ domains.

In most industrialized democracies, there is a legal and ethical mandate to provide work that is safe from psychological and physical harm [2]. To do this, organizations will require instruments capable of identifying psychosocial hazards and assessing the level of risk they pose to workers. Every workplace is unique and will require a psychosocial hazard and risk approach tailored and unique to their organizational context [6,7]. Identifying an instrument that provides adequate coverage of context-specific hazards and considers worker socio-demographics, occupation, and type of workplace, will be the most relevant and informative [5,132]. To this end, summaries of these psychosocial hazard instruments have been published, aiming to identify their psychometric quality [134,135], how relevant they are in assessing a theoretical model [132], their limitations [23,136,137], and identifying appropriateness to organizational context [132,138,139,140,141]. Most of the ‘prevent harm’ relevant instruments that were located through this review collect data from workers and may be the most suitable for the ‘monitor, review, and improve’ activity through aggregate and subgroup data analysis. Instruments that are suitable for the ‘identify gaps and opportunities’ and ‘identify priorities and design new/enhanced interventions’ were less common but may be suitable for some organizations [50,91,92,100,103,104]. Most of the instruments relevant to the ‘prevent harm’ domain had resources or information available regarding their development.

To complement psychosocial hazard reduction strategies, under an integrated approach, organizations should focus on strength-based approaches to maximize positive mental health among workers [6,7]. Instruments relevant to the ‘promote the positive’ domain were far less common in this review and primarily focused on the ‘monitor, review, and improve’ activity, aimed at individual-level employee well-being factors. The individualized focus of positive measures has previously been observed and may in part be explained by the fact that positive psychology is a newer discipline [6,7,16,17]. However, some instruments were designed to assess organizational level factors, that could improve positive mental health in workers, such as social capital and meaningful work [107,115,118]. The emergence of a systems informed positive psychology, however, provides a solid foundation for the evolution of more organizational level instruments [142]. No instruments were located that may be suitable for the ‘identify gaps and opportunities’ and/or ‘identify priorities and design new/enhanced interventions’ activities in relation to the ‘promote the positive’ domain. Most of the instruments relevant to the ‘promote the positive’ domain had resources or information available regarding their development.

Workplace mental health instruments for assessing the ‘respond to problems’ domain were the least common in this review. Nine of the thirteen instruments relevant to the ‘respond to problems’ domain may be suitable for the ‘identify gaps and opportunities’ and/or ‘identify priorities and design new/enhanced interventions’ activities. The nine instruments located in this review may assist an organization seeking to audit their current services, practices, and programs. Most of these instruments focused on auditing the presence of mental health literacy training, mental health leadership and access and awareness of mental health services [33,121,122,123,124,125,126,128,129,130]. There were limited instruments that may be suitable for the ‘monitor, review, and improve’ activity, with those available designed for assessing factors related to obstacles for return to work, mental health literacy and mental illness stigma [122,123,126]. Many of the ‘respond to problems’ instruments located in this review lacked resources and information on how the instrument was developed. A lack of instruments that provided development resources was also found in a previous rapid review of recovery at work instruments that was unable to identify any formal evaluation of the instruments that they located [143].

When determining which instruments to use to implement an integrated approach to workplace mental health, organizations should ensure that those that they select have information regarding the instrument’s development. In Australia, People at Work is one example of a free psychosocial risk assessment, that has information on how the instrument was developed due to the collaboration between researchers, practitioners, and decision-makers [70]. Similar collaborations can also be seen in the UK, where the Health and Safety Executive created readily available and free instruments to help employers assess their organizations against the Management Standards [49,59]. The HSE Stress Indicator and Line Manager Competency instruments are two examples of instruments that have researcher-developed information and resources [49,59]. In the United States, the Generic Job Stress Questionnaire and Worker Well-Being Questionnaire, developed by the National Institute for Occupational Safety and Health, are also readily available, free for employer use, and have resources on how these instruments were developed [27,63].

### Limitations and Recommendations for Future Research

Due to the purpose of this scoping review, it was necessary to focus on the core components of each domain of an integrated approach to avoid overcomplicating our coding strategy. It is recommended that organizations that are interested in implementing an integrated approach still seek guidance from experts. Some guidance materials, in the form of journal articles, fact sheets, books, and commercial publishing houses have been created to help organizations and researchers locate instruments, however, a free and centralized platform for housing these instruments would be an efficient way of assisting organizations to locate suitable instruments for their organizational context.

This scoping review only included instruments produced in the English language, with eight non-English (Japanese, Korean, French, Italian, Dutch) instruments excluded. Due to this language restriction the identified instruments were predominately developed in the U.S.A., Australia, Canada, and Europe. Further research into instruments published in other languages, and those developed in other countries, is needed to broaden the scope of understanding of how other instruments can be utilized to assist organizations wishing to implement an integrated approach to workplace mental health. We encourage practitioners to consider instruments catalogued in our review as they progress towards implementing an integrated approach to workplace mental health. We also recommend that researchers continue to work on the creation and development of new instruments that can assist organizations with these efforts, particularly in the domains of ‘promote the positive’ and ‘respond to problems’, and all domains collectively.

## 5. Conclusions

This paper located 109 instruments that organizations can use when implementing an integrated approach to workplace mental health. There were some instruments located that may be relevant to multiple domains of an integrated approach. Most of the instruments located, though, were relevant to the ‘prevent harm’ domain, primarily suitable for the ‘monitor, review and improve’ KTA activity. Some instruments were relevant to the ‘promote the positive’ domain, with most suitable for the ‘monitor, review and improve’ KTA activity, that look at individual, rather than job or workplace, factors. Limited instruments relevant to the ‘respond to problems’ domain were located, with most being suitable for the ‘identify gaps and opportunities’ and/or ‘identify priorities and design new/enhanced interventions’ KTA activities. Most instruments had development information and resources. There is an opportunity for the expansion of instruments available that more fully reflect the variety of workplace supports and strategies in relation to the ‘promote the positive’ and ‘respond to problems’ domains and an integrated approach in its entirety.

## Figures and Tables

**Figure 1 ijerph-20-01192-f001:**
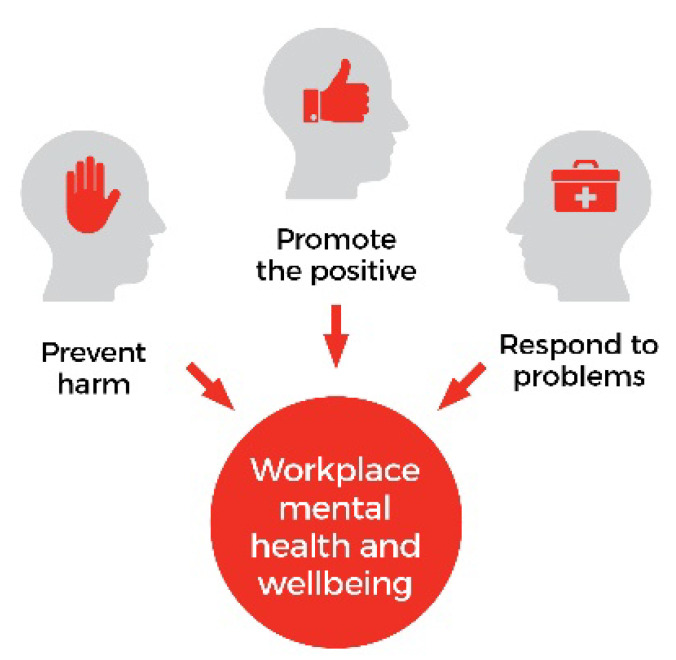
An integrated approach to workplace mental health.

**Figure 2 ijerph-20-01192-f002:**
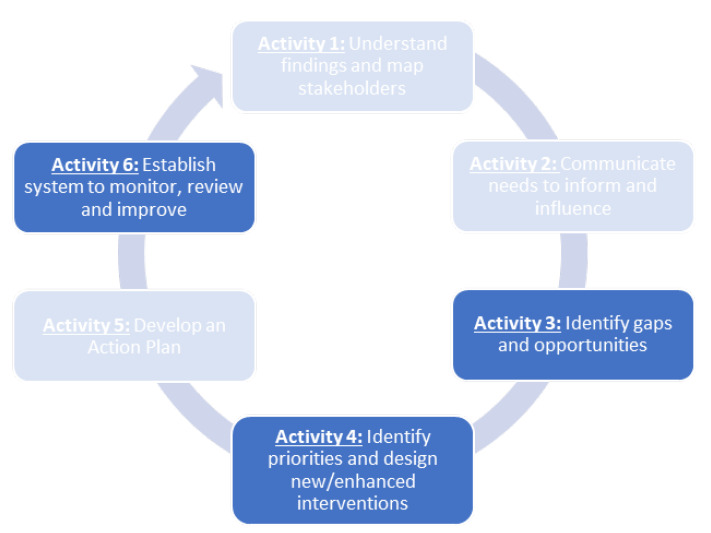
Adapted knowledge to action (KTA) framework.

**Figure 3 ijerph-20-01192-f003:**
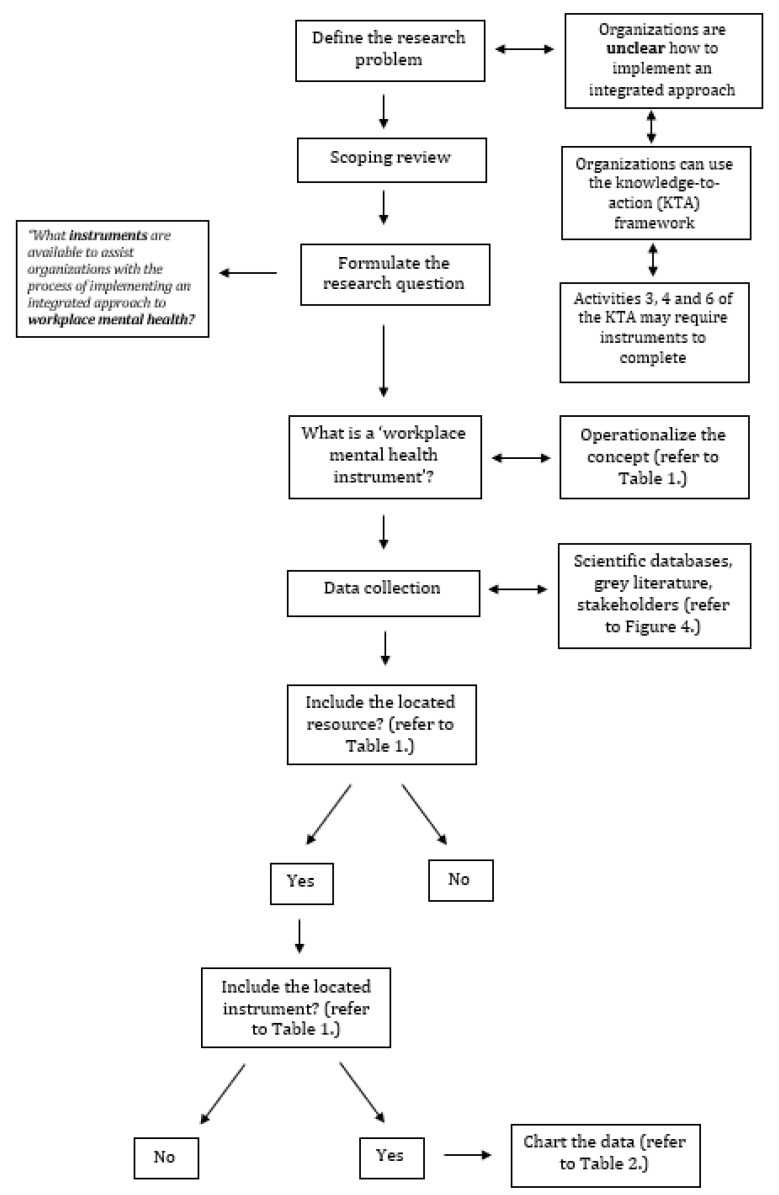
Methods flowchart.

**Figure 4 ijerph-20-01192-f004:**
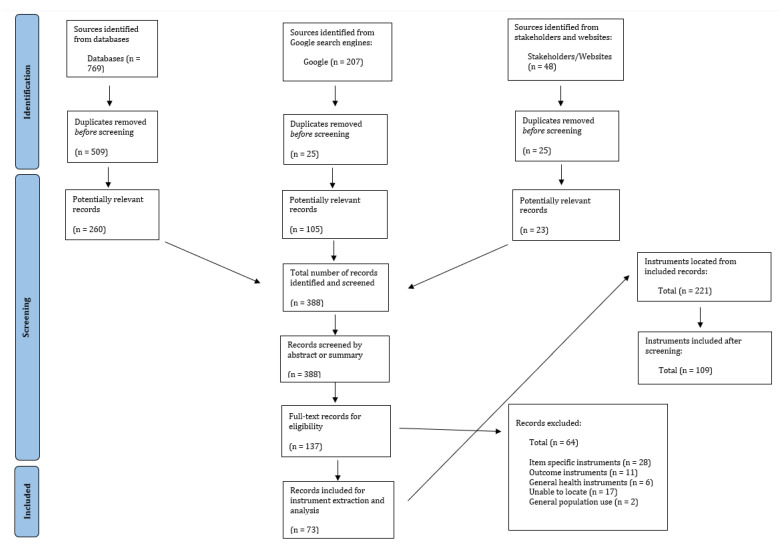
PRISMA flow diagram.

**Table 1 ijerph-20-01192-t001:** Eligibility criteria.

*Inclusion Criteria:*	*Exclusion Criteria:*
(a) workplace mental health instruments ^1^ (b) produced in the English language (c) instruments that assessed more than one work-related mental health factor ^2^ (d) instruments with resources available for review	(a) instruments that could only be accessed through purchase (b) general health instruments (c) instruments created for general population use or annual surveys (d) individual outcome instruments ^3^ (e) factor, job sector or country specific instruments

^1^ For the purposes of this review, a “workplace mental health instrument” needs to be both (i) work-specific, identified as an instrument that either includes “work”, “employee” or “occupational” in the title or with scales or items that refer to work, and (ii) measure at least one of the following variables related to the different domains of “an integrated approach to workplace mental health”: *prevent harm* (“psychosocial hazards” or “psychosocial risk assessment”), *promote the positive* (“well-being”, “character strengths”, “employee engagement”, “psychological capital”, “social capital”, “organizational climate”, “meaningful work”, “positive psychosocial factors”, “positive psychological assessment”), *respond to problems* (“psychoeducation”, “mental health stigma”, “return to work after a mental health problem”, “help seeking behavior”, “mental health literacy”). ^2^ An integrated approach to workplace mental health acknowledges that external to work factors also need to be assessed and managed, however, for the purposes of limiting the scope of this paper only those instruments that assess work-related factors are considered. ^3^ Outcome instruments are defined as those that assessed an individual’s mental health status, such as the Kessler-10 assessment of psychological distress or the Beck’s depression inventory.

**Table 2 ijerph-20-01192-t002:** Definitions used for mapping instruments to the integrated approach domains and knowledge to action (KTA) activities.

Coding Categories	Mapping Definitions
**Alignment with integrated approach domains: determining whether an instrument can assess ‘prevent harm’, ‘promote the positive’, or ‘respond to problems’**	An instrument was categorized as being the most relevant for assessing one or all of the domains of an integrated approach to workplace mental health if it assessed ONE or more of the below variables: *Prevent harm*: “psychosocial hazards” or “psychosocial risk assessment”; *Promote the positive*: “well-being”, “character strengths”, “employee engagement”, “psychological capital”, “social capital”, “organizational climate”, “meaningful work”, “positive psychosocial factors”, “positive psychological assessment”; *Respond to problems*: “psychoeducation”, “mental health stigma”, “return to work after a mental health problem”, “help seeking behavior”, “mental health literacy”.
**KTA activities: determining whether an instrument may be suitable to assist organizations to ‘identify gaps and opportunities’, ‘identify priorities and design new/enhanced interventions’, ‘monitor, review and improve’ or for multiple purposes**	*Identify gaps and opportunities*: instruments for auditing the current services, practices, and programs currently available to an organization; *Example of a service, practice, or program*: specialist psychological/psychiatry services, mental health literacy training or employee assistance programs (EAP) *; *Identify priorities and design new/enhanced interventions*: instruments for identifying opportunities to enhance the current services, practices, and programs currently available to an organization; *Monitor, review and improve*: instruments for reviewing process implementation and outcome effectiveness. * Examples taken from Beyond Blue knowledge translation guide.
**Accessibility of information about instruments: resources/information on the development of an instrument**	Instruments were classified as having information on development based on the absence or presence of resources available. A classification was given if there were resources available, whether in the form of a research paper, grey literature source or website that explained how the instrument was developed. This could include how the instrument was tested for reliability and/or validity, however, it was beyond the remit of this paper to assess psychometrics in more detail. Information or resources were sourced through scientific databases, grey literature, and stakeholder expertise.

**Table 3 ijerph-20-01192-t003:** List of instruments.

**Integrated Approach (‘Prevent Harm’, ‘Promote the Positive’, ‘Respond to Problems’)**			
**Workplace Mental Health Instrument**	**Author and Year**	**Country of Origin**	**Brief Description of Instrument**
FlourishDX	People Diagnostix 2022 [25]	Australia	A commercially designed customizable suite of resources, including a work design survey, for assessing psychosocial factors at work
Guarding Minds at Work	Workplace Strategies for Mental Health 2009 [26]	Canada	A suite of resources for assessing psychological health and safety in the workplace
**‘Prevent Harm’ and ‘Promote the Positive’**			
**Workplace Mental Health Instrument**	**Author and Year**	**Country of Origin**	**Brief Description of Instrument**
National Institute for Occupational Safety and Health (NIOSH) Worker Well-Being Questionnaire	Chari et al., 2021 [27]	United States	68 item questionnaire for assessing worker well-being
Quality of Work Life Measure (QWL)	Sirgy et al., 2001 [28]	United States	80 item measure for assessing the work environment, job requirements, supervisor behavior and ancillary programs
Scale for Daily Hassles and Uplifts at Work	Junca-Silva et al., 2020 [29]	Portugal	50 item scale for measuring workplace hassles and uplifts
The Basic Psychological Need Satisfaction and Need Frustration at Work Scale	Chen et al., 2015 [30]	Norway	24 item scale for assessing need satisfaction and frustration at work
Work Experience Measurement Scale (WEMS)	Nilsson et al., 2013 [31]	Sweden	32 item scale for assessing the experiences of work- and work-related situations from a salutogenic perspective
Workplace Psychological Safety Index (WPSI)	AP Psychology and Consulting Services, 2022 [32]	Australia	123 item commercially designed questionnaire for assessing psychosocial risk factors at work
**‘Prevent Harm’ and ‘Respond to Problems’**			
**Workplace Mental Health Instrument**	**Author and Year**	**Country of Origin**	**Brief Description of Instrument**
Healthy Workplace Audit Tool	WorkSafe Queensland (QLD), 2020 [33]	Australia	42 item tool for assessing workplace systems and environments
Mental Health Audit	People Diagnostix, 2022 [34]	Australia	14 item tool for auditing an organization’s available resources and supports for employee mental health
**‘Prevent Harm’**			
**Workplace Mental Health Instrument**	**Author and Year**	**Country of Origin**	**Brief Description of Instrument**
Areas of Worklife Scale	Leiter and Maslach, 2003 [35]	Australia	29 item scale for assessing organizational life
A Shortened Stress Evaluation Tool (ASSET)	Johnson, 2008 [36]	UK	37 item questionnaire for screening employee stress
Australian Workplace Barometer (AWB)	Dollard et al., 2012 [37]	Australia	126 item surveillance tool used to monitor psychosocial risks in the workplace
BHF Sample Audit Tool	British Heart Foundation, 2017 [38]	UK	45 item auditing template for workplace health
Brief Job Stress Questionnaire (BJSQ)	Inoue et al., 2014 [39]	Japan	A questionnaire developed in two different lengths (80 item—short version, 141 item—standard version) for measuring psychosocial factors at work
Copenhagen Psychosocial Questionnaire (COPSOQ)	COPSOQ III. Guidelines and Questionnaire, 2019 [40]	Denmark	A questionnaire developed in three different lengths (32 item—core version, 60 item—middle version, 152 item—long version) for assessing psychosocial factors at work
CDC NHWP Health and Safety Climate Survey (INPUTS)	Centre for Disease Control, 2011 [41]	United States	23 item instrument for measuring workplace characteristics associated with employee health outcomes and injury rates
Danish Psychosocial Work Environment Questionnaire	Clausen et al., 2019 [42]	Denmark	119 item instrument for assessing psychosocial working conditions
Decent Work Scale	Duffy et al., 2017 [43]	United States	15 item scale for assessing the attainment of decent work among employed adults
Demand-Induced Strain Questionnaire (DISQ)	Bova et al., 2013 [44]	The Netherlands	31 item instrument for measuring job demands and job resources
Effort-Reward Imbalance Questionnaire (ERI)	Siegrist et al., 2004 [45]	Germany	A questionnaire developed in two different lengths (16 item—short version, 23 item—original version) for assessing perceived demands and rewards at work
Group Nurturance Inventory (GNI)	Johansson and Biglan, 2021 [46]	Norway	17 item behavioral assessment instrument intended for use with workgroups
Healthy Workplace All Employee Survey	The Center for the Promotion of Health in the New England Workplace (CPH-NEW), 2014 [47]	United States	36 item survey designed to assess employee attitudes related to health, safety, and wellness
Healthy Work Survey	Centre for Social Epidemiology, 2022 [48]	United States	A survey developed in two different lengths (90 item—short version, 116 item—long version) for measuring work stressors
Health and Safety Executive (HSE) Indicator Tool	Edwards et al., 2008 [49]	UK	35 item measure for assessing workplace stress
ILO Stress Checkpoints	International Labor Organization, 2012 [50]	Multiple countries	50 item checklist for reviewing workplace conditions that may lead to employee stress
Index of Psychological Well-Being at Work	Dagenais-Desmarais and Savoie, 2012 [51]	Canada	25 item index for assessing psychological well-being at work
iWorkHealth	Abdin et al., 2019 [52]	Singapore	27 item psychosocial health assessment tool for identifying common workplace stressors
Job Characteristics Index (JCI)	Sims et al., 1976 [53]	United States	30 item instrument for assessing job characteristics and employee attitudes and behavior
Job Content Questionnaire (JCQ)	Karasek et al., 1988 [54]	United States	A tool developed in three different lengths (39 item—JCQ2, 49 item—JCQ, 79 item—JCQ2 researcher version) for psychosocial job assessment
Job Diagnostics Survey (JDS)	Hackman and Oldham, 1975 [55]	United States	A survey developed in three different lengths (15 item—revised short version, 53 item—short version, 83 item—full version) for assessing work motivation and job redesign
Job Related Tension Index	Kahn et al., 1964 [56]	United States	15 item index for examining job-related tension
Job Stress Survey (JSS)	Vagg and Spielberger, 1999 [57]	United States	30 item survey for measuring occupational stress
APHIRM (A Participative Hazard Identification and Risk Management) Toolkit	Oakman and MacDonald, 2019 [58]	Australia	48 item workplace hazard identification and risk management tool
Line Manager Competency Tool	Toderi et al., 2016 [59]	UK	A tool developed in two different lengths (36 item—brief version, 66 item—full version) for assessing management behaviors for preventing and reducing stress at work
Measure of Psychosocial Risk Factors and Burnout	Jacobo-Galicia and Maynez-Guaderrama, 2020 [60]	Mexico	80 item scale for evaluating psychosocial risk factors and burnout in the workplace
Multimethod Job Design Questionnaire (MJDQ)	Campion and Thayer, 1985 [61]	United States	A questionnaire developed in two different lengths (48 item—revised version, 70 item—original version) for assessing job design
New Organizational Role Stress Scale (NORS)	Srivastav, 2009 [62]	India	71 item scale for assessing role stress at work
NIOSH Generic Job Stress Questionnaire	Hurrell and McLaney, 1988 [63]	United States	246 item questionnaire for assessing job stress
Occupational Resilience Assets Questionnaire (ORA-Q)	Magrin et al., 2017 [64]	Italy	18 item questionnaire for assessing resilience resources at work
Occupational Stress Index (OSI)	Belkic and Savic, 2008 [65]	United States	65 item index for assessing key modifiable work stressors
Occupational Stress Indicator (OSIND)	Robertson et al., 1990 [66]	UK	A scale developed in three different lengths (94 item—abridged version, 167 item—original version, 188 item—revised version) for assessing job satisfaction, mental health, and type A behavior
Organizational Climate Scale (CLIOR)	Suarez et al., 2012 [67]	Spain	A scale developed in two different lengths (15 item—brief version, 50 item—full version) for assessing organizational climate
Organizational Justice Scale	Niehoff and Moorman, 1993 [68]	United States	20 item scale for assessing organizational justice
OrgFit	Jiminez and Dunkl, 2017 [69]	Germany	56 item instrument for assessing psychosocial risks at work
People at Work Survey	Jimmieson et al., 2016 [70]	Australia	103 item psychosocial risk assessment survey
Perceived Work Characteristics Survey	Haynes et al., 1999 [71]	UK	41 item survey for assessing the psychological well-being of employees
Pressure Management Indicator (PMI)	Malkiewicz et al., 2016 [72]	UK	146 item questionnaire for measuring occupational stress
Psychologically Safe Leader Assessment	Workplace Strategiesfor Mental Health, 2016 [73]	Canada	65 item survey for assessing positive leadership strategies that promote psychological health and safety at work
PsyHealth	Kuczynski et al., 2020 [74]	Germany	33 item questionnaire for assessing psychosocial work conditions
Psychosocial Safety Climate (PSC-12) Survey	Hall et al., 2010 [75]	Australia	12 item instrument for measuring psychosocial safety climate
Psychosocial Working Conditions (PWC)	Widerszal-Bazyl and Cieslak, 2000 [76]	Poland	36 item instrument for monitoring stress at work
General Nordic Questionnaire for Psychological and Social Factors at Work (QPSNORDIC)	Dallner et al., 2000 [77]	Denmark	A questionnaire developed in two different lengths (34 item—short version, 123 item—long version) for assessing psychological and social factors at work
Quality of Worklife Module Questionnaire	NIOSH, 2013 [78]	United States	70 item questionnaire for measuring work life and work experience
Quality of Worklife Scale (WRQoL)	Easton and Van Laar, 2012 [79]	UK	24 item measure for assessing the quality of working life
Questionnaire on the Experience and Evaluation of Work (VBBA)	Veldhoven et al., 2015 [80]	The Netherlands	A questionnaire developed in two different lengths (108 item—abridged version, 232 item—full version) for assessing work, well-being, and performance
Role Ambiguity/Conflict Scales	Rizzo et al., 1970 [81]	United States	30 item scale for measuring role ambiguity and role conflict at work
Role Hassles Index (RHI)	Zohar, 1997 [82]	Israel	20 item index for measuring job demands
SHAPE (The Survey for Health, Attendance, Productivity and Engagement)	SHAPE Global Ltd., 2022 [83]	Australia and UK	237 item commercially designed survey for measuring employee productivity
START procedure	Satzer and Geray, 2009 [84]	Germany	41 item mental stress risk assessment at work
Stress Diagnostics Survey (SDS)	Ivancevich and Matteson, 1988 [85]	United States	80 item survey for measuring job related stress
Stress Profile	Setterlind and Larsson, 1995 [86]	Sweden	224 item instrument for measuring stress in life and work at the individual, group, and organizational level
Stress Satisfaction Offset Score (SSOS)	Shain, 2021 [87]	Canada	4 item survey for assessing risks to mental and physical health at work
Structured Multidisciplinary Work Evaluation Tool (SMET Questionnaire)	Haraldsson et al., 2019 [88]	Sweden	30 item questionnaire for evaluating work environmental interventions
Survey of Organizational Characteristics (SOC)	Thumin and Thumin, 2011 [89]	United States	83 item instrument for assessing organizational climate
Swedish Demand Control Support Questionnaire (DCSQ)	Sanne et al., 2005 [90]	Norway	17 item psychosocial job assessment questionnaire
The Standard Audit Tool	Workplace Strategiesfor Mental Health, 2013 [91]	Canada	62 item tool for auditing the Canadian national standard for psychological health and safety in the workplace
WA Government—Mines, Industry and Safety Risk Assessment Tool	Government of Western Australia, 2014 [92]	Australia	23 item risk assessment tool for assessing risks to psychological injury at work
Well-Being Inventory	Vendrig et al., 2018 [93]	The Netherlands	82 item tool for screening employees for risk factors for prolonged or future sickness absence
Working Conditions and Control Questionnaire (WOCCQ)	Hansez, 2008 [94]	Belgium	80 item job control and stress questionnaire
Work Design Questionnaire (WDQ)	Morgeson and Humphrey, 2006 [95]	United States	77 item questionnaire for measuring work characteristics
Work Environment Scale (WES)	Moos, 1981 [96]	United States	A scale developed in two different lengths (40 item—short version, 90 item—long version) for measuring social environments at work
Work-Health-Check (WHC)	Gadinger et al., 2012 [97]	Germany	42 item questionnaire for assessing work-related psychosocial stress
Work Organization Assessment Questionnaire (WOAQ)	Griffiths et al., 2006 [98]	UK	28 item questionnaire for assessing psychosocial hazards at work
Workplace Integrated Safety and Health (WISH) Assessment	Sorensen et al., 2018 [99]	United States	40 item tool for measuring workplace practices that can determine worker safety and health outcomes
Workplace Pulse Check	SafeWork New South Wales (NSW), 2021 [100]	Australia	11 item tool for auditing actions that can contribute to a mentally healthy workplace
Workplace Stressors Assessment Questionnaire (WSAQ)	Mahmood et al., 2010 [101]	United States	22 item questionnaire for assessing stress related factors at work
Workplace Stressors Questionnaire (WSQ)	Holmgren et al., 2009 [102]	Sweden	20 item questionnaire for screening at risk employees impacted by work-related stress
WorkPlace Wellbeing Insights Survey	WorkWell Technical Report, 2021 [103]	Australia	148 item survey for assessing workplace health and safety
Work-Related Stress Questionnaire (WRSQ)	De Sio et al., 2020 [104]	Italy	33 item questionnaire for assessing psychosocial risks at work
Worksafe Queensland Psychosocial Risk Assessment	WorkSafe QLD, 2010 [105]	Australia	26 item tool for assessing psychosocial risks at work
WorkWell Index	Mauss et al., 2017 [106]	Germany	10 item questionnaire for assessing work-related stress
**‘Promote the positive’**			
**Workplace Mental Health Instrument**	**Author and Year**	**Country of Origin**	**Brief Description of Instrument**
Comprehensive Meaningful Work Scale (CMWS)	Lips-Wiersma and Wright, 2012 [107]	New Zealand	28 item scale for measuring meaningful work
Employee Well-Being Scale	Pradhan et al., 2019 [108]	India	31 item scale for assessing employee well-being at work
Eudaimonic Workplace Well-Being Scale	Bartels et al., 2019 [109]	United States	8 item scale for measuring eudaimonic workplace well-being
Mental Fitness and Resiliency Inventory (MFRI)	Peterson et al., 2020 [110]	Canada	32 item tool for measuring the presence of positive practices that contribute to healthy and effective workplace cultures
New Measure of Employee Engagement	Ababneh et al., 2019 [111]	New Zealand	20 item measure for assessing employee engagement
Psychological Capital Questionnaire	Luthans et al., 2007 [112]	United States	24 item questionnaire for assessing psychological capital at work
Psychological Empowerment in the Workplace Scale	Spreitzer, 1995 [113]	United States	12 item scale for assessing psychological empowerment at work
R.I.G.H.T Leadership Scale	Gulseren et al., 2021 [114]	Canada	15 item scale for measuring psychologically healthy leadership at work
The Work as Meaning Inventory (WAMI)	Steger et al., 2012 [115]	United States	10 item scale for measuring meaningful work
Work-Related Well-Being Index (WRWB)	Eaton et al., 2018 [116]	United States	11 item instrument for assessing worker well-being
Workplace PERMA Profiler	Kern, 2014 [117]	Australia	23 item questionnaire for measuring positive emotion, engagement, relationships, meaning, and accomplishment (PERMA) in the context of work
Workplace Social Capital Scale	Kouvonen et al., 2006 [118]	UK	8 item scale for measuring social capital at work
Workplace Well-Being Questionnaire (WWQ)	Parker and Hyett, 2011 [119]	Australia	31 item measure for assessing workplace well-being
Belonging Index	Making Work Absolutely Human (MWAH), 2021 [120]	Australia	35 item commercially designed index for assessing whether workers have a sense of belonging
**‘Respond to problems’**			
**Workplace Mental Health Instrument**	**Author and Year**	**Country of Origin**	**Brief Description of Instrument**
Employee Quiz to Assess the Current State of Mental Healthcare Accessibility in Your Organization	National Alliance on Mental Illness (NAMI), 2020 [121]	United States	20 item survey for assessing an organization’s current approach to mental health care and employee perceptions of current services offered
Employee Survey to Assess Your Team’s Knowledge and Feelings About Mental Health	National Alliance on Mental Illness (NAMI), 2020 [121]	United States	20 item survey for assessing employee knowledge, understanding, and feelings about key workplace mental health issues
Manager’s Attitude Towards Depression Survey Measure	Martin, 2010 [122]	Australia	38 item survey for measuring manager attitudes toward depressed employees
Mental Health Literacy Workplace Scale (MHL-W)	Moll et al., 2017 [123]	Canada	16 item scale for assessing mental health literacy in the workplace
Mental Health Visibility and Accessibility Checklist	National Alliance on Mental Illness (NAMI), 2020 [121]	United States	11 item checklist for assessing the visibility and accessibility of employee mental health resources
Mentally Healthier Workplaces Self-Assessment Tool	ACT Government, 2017 [124]	Australia	39 item survey for measuring the capability of workplaces to create a mentally healthy workplace
NSW Benchmarking Tool	Donnelly and Lewis 2017 [125]	Australia	42 item survey for measuring the capability of workplaces to create a mentally healthy workplace
Return-to-Work Obstacles and Self-Efficacy Scale (ROSES)	Corbiere et al., 2017 [126]	Canada	46 item scale for assessing return to work obstacles and employee self-efficacy
The Workplace Scale (WPS)	McHugh, 2016 [127]	Canada	31 item assessment tool for evaluating employee perceptions and knowledge of workplace practices and procedures that can develop healthy, supportive workplaces
Vitality’s Health Metric Scorecard	Vitality Institute, 2015 [128]	United States	A scorecard developed in two different lengths (10 item—core version, 73 item—comprehensive version) for capturing workforce health metrics
WorkCover QLD Organizational Benchmarking Tool	Queensland Government, 2017 [129]	Australia	42 item tool for assessing whether an organizational has a systematic approach to work health and safety, worker health and well-being and worker’s compensation and return to work
Workplace Mental Health Assessment	American Psychiatric Association (APA), 2019 [130]	United States	20 item assessment of workplace mental health resources
Workplace Mental Health Assessment Survey	Black Dog Institute, 2021 [131]	Australia	29 item commercially designed survey for assessing workplace mental health

**Table 4 ijerph-20-01192-t004:** Mapping of instruments to integrated approach domains and knowledge to action (KTA) activities.

	‘Prevent Harm’	‘Promote the Positive’	‘Respond to Problems’	‘Prevent Harm’ and ‘Promote the Positive’	‘Prevent Harm’ and ‘Respond to Problems’	All Domains	Total
**Integrated approach to workplace mental health domains**	**72**	**14**	**13**	**6**	**2**	**2**	**109**
	**‘Identify gaps and opportunities’/‘Identify priorities and design new/enhanced interventions’**		**‘Monitor, review and improve’**		**‘All activities’**		**Total**
**KTA framework activities ***	**15**		**90**		**4**		**19**

* See Table 2 for how these activities were defined.

## Data Availability

The data presented in this study are available on request from the corresponding author.

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
