# Peer review of "An Integrated Approach to Workplace Mental Health: A Scoping Review of Instruments That Can Assist Organizations with Implementation"

_ijerph, 2023, doi:10.3390/ijerph20021192_

Round 1

Reviewer 1 Report

The theme is interesting and brings a contribution to the area. It would be interesting to have a discussion about the differences and points in common in relation to the instruments allocated in each category. For example, in the category, 'Respond to problems', what problems do they set out to answer? What still needs to be developed? I suggest reorganizing Table 4 based on this logic.

Author Response

We thank the reviewer for this thoughtful feedback. The goal of this section and Table 4 was to identify how many instruments were aligned with each domain of an integrated approach to workplace mental health. The intention was to highlight to the reader that most of the instruments were mapped to the ‘prevent harm’ domain and the ‘monitor, review, and improve’ KTA activity. The focus of the review was on how the instruments aligned to the different domains of an integrated approach and less about the specific instruments categorized under each domain. For greater clarity we have updated the methods section (page 8, in the paragraph below Table 2) to explain how Table’s 3 and 4 were designed to answer our research question. In addition, to provide more information to readers, we have given examples in our results (page 23, in the section titled “alignment with integrated…”) of the differences and similarities among the instruments allocated to the categories in the table.

Reviewer 2 Report

Excellent paper that offers a significant contribution to the field of mental health in the workplace at all stakeholder levels. The work is clear, comprehensive, and relevant for the field and presented in a well-structured manner. I recommend that the manuscript be accepted in its current form.

The scoping review aimed to identify instruments that may assist organizations in implementing an integrated approach to workplace mental health using the knowledge to action (KTA) framework. More specifically, the authors identified work-specific instruments that were relevant to three overlapping domains of an integrated approach to workplace mental health: prevent harm, promote the positive and respond to problems, as well as support three specific KTA activities: identifying gaps and opportunities, identifying priorities, and designing new/enhanced interventions, and to monitor, review and improve.  

The topic is original and relevant to the field because an integrated approach to workplace mental health is still an emerging area of research. This scoping review adds to the current literature by exploring instruments that are available to assist organizations with the process of implementing an integrated approach to workplace mental health. 

The research is scientifically sound, and the methodology is appropriate. The scoping review was conducted according to the PRISMA-ScR statement, which is appropriate for the aim of the scoping review. The eligibility criteria of the reviewed literature, as well as the information sources and search, selection of sources of evidence, data charting process and coding categories are clearly presented and appropriate.

The authors were able to clearly establish whether the collected instruments could assess “prevent harm”, “promote the positive”, and/or “respond to problems, whether the instruments could be suitable to assist organizations to “identify gaps and opportunities”, “identify priorities and design new/enhanced interventions” and/or “monitor, review and improve”. Furthermore, the accessibility of information about instruments themselves is also presented, which will help organizations choose the best instrument for their needs. The research limitations and recommendations for future research are also relevant and clearly identified by the authors.

The tables and figures support the reader’s understanding by offering synthetized and relevant information on the integrated approach to workplace mental health, the knowledge to action (KTA) framework, the methods flowchart, the eligibility criteria, the definition used for mapping instruments, the PRISMA flow diagram, and the list of instruments categorized by the three domains of an integrated approach to workplace mental health.

The conclusions are consistent with the evidence and arguments presented and clearly address the main study question. The cited references are appropriate and include relevant publications for the scoping review.

Author Response

We thank the reviewer for their positive feedback.

Reviewer 3 Report

Dear Authors,

I don't know if the article is that interesting. The article is definitely too long.

It contains a lot of information. The article should be improved to make it better to read.

The introduction must be shortened.

Figures are too small, for example figures 1, 3, . Fugure 1, 3 are not visible. I can't read it.

Table 3 should be corrected. It lacks information such as:

- type of article (paper article? or literature review?)

- what does "items" mean?

- in which database was the article found?

- a brief description of the individual instruments and study?

Discussion

- each paragraph should have a reference.

References - please improve.

Author Response

Comment 1: I don't know if the article is that interesting. The article is definitely too long.

Response 1: We thank the reviewer for their feedback. We have shortened the size of the introduction and discussion, aligning both sections with the average word length in other IJERPH manuscripts.

Comment 2: It contains a lot of information. The article should be improved to make it better to read

Response 2: We have shortened the size of the introduction, specifically removing any duplicated or superfluous information. We have condensed the overview of an integrated approach section (page 2), removed unnecessary information from our eligibility criteria (page 5) and added more clarity around how our tables were designed to answer our research question (page 8). We have updated Table 3 to be more readable (pages 9-21) and have removed unnecessary information from our discussion (pages 23-24) and limitations section (page 24).

Comment 3: The introduction must be shortened.

Response 3: We have shortened the size of the introduction.

Comment 4: Figures are too small, for example figures 1, 3, . Fugure 1, 3 are not visible. I can't read it.

Response 4: We have increased the size of Figures 1 (page 2) and 3 (page 3).

Comment 5: Table 3 should be corrected. It lacks information such as:

- type of article (paper article? or literature review?)

- what does "items" mean?

- in which database was the article found?

- a brief description of the individual instruments and study?

Response 5: We have added a sentence to our methods section to clarify the meaning of “items” (page 4). We appreciate the reviewer’s efforts to further enhance Table 3 with the addition of information on the “type of article” and “database where the article was found”. The addition of this information would be appropriate and helpful in a standard scoping review of articles; however, our scoping review was of instruments. The articles used to locate the instruments were also sourced across multiple databases, with many crossovers observed as data saturation was reached, we are concerned that adding this information may induce confusion rather than clarity. However, we have amended the table by removing the “number of items” section and adding a “brief description of instrument” section (pages 9-21). We thank the reviewer for the valuable feedback regarding Table 3 as the extra information will be very helpful for our readers.

Comment 6: Discussion

- each paragraph should have a reference.

Response 6: We appreciate the reviewer’s efforts to enhance our discussion. Our discussion does contain references in each paragraph, aside from the opening paragraph that was intended as a summary of our results. The opening summary was then contextualized within the literature in the following paragraphs. This approach appears to be consistent with some other scoping reviews published in IJERPH.

Comment 7: References - please improve.

Response 7: We thank the reviewer for this feedback and have made significant changes to our references. We have improved our references using the “Reference List and Citations Style Guide for MDPI Journals”. The layout of the references is now more readable and consistent (pages 25-33).

Round 2

Reviewer 3 Report

Dear Authors,

Thank You the Authors for improving the work.
Currently, the manuscript is very good. I recommend the manuscript to publication in the current form.

Sincerely,